

# Superdiffusive transport in quasi-particle dephasing models

Yu-Peng Wang[1,2], Chen Fang[1,3] and Jie Ren[1,2,4*]

**1** Beijing National Laboratory for Condensed Matter Physics and Institute of Physics,
Chinese Academy of Sciences, Beijing 100190, China
**2** University of Chinese Academy of Sciences, Beijing 100049, China
**3** Kavli Institute for Theoretical Sciences,
Chinese Academy of Sciences, Beijing 100190, China
**4** School of Physics and Astronomy, University of Leeds, Leeds LS2 9JT, United Kingdom

★ jieren@iphy.ac.cn

## Abstract

Investigating the behavior of noninteracting fermions subjected to local dephasing, we reveal that quasi-particle dephasing can induce superdiffusive transport. This superdiffusion arises from nodal points within the momentum distribution of local dephasing quasi-particles, leading to asymptotic long-lived modes. By studying the dynamics of the Wigner function, we rigorously elucidate how the dynamics of these enduring modes give rise to Lévy walk processes, a renowned mechanism underlying superdiffusion phenomena. Our research demonstrates the controllability of dynamical scaling exponents by selecting quasi-particles and extends its applicability to higher dimensions, underlining the pervasive nature of superdiffusion in dephasing models.

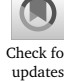
## 1  Introduction

Transport properties of particles, energy, and information in nonequilibrium quantum many-body systems have garnered significant attention [1–12]. The emergence of anomalous transport, which deviates from the classical diffusion characterized by linear growth in mean square displacement over time, challenges established principles in quantum many-body dynamics. This anomaly includes superdiffusion [13] and subdiffusion [14], where particle spreading occurs faster or slower than classical expectations.

Notably, the one-dimensional Heisenberg model has exhibited superdiffusion [15–22] with a dynamical exponent of $z = 3/2$ and a scaling function within the Kardar-Parisi-Zhang (KPZ) universality class [23–25]. This superdiffusive behavior extends to a broader class of integrable models with non-Abelian symmetries [26–31], with transitions to diffusive behavior observed when integrability or symmetry is perturbed [32–35]. Superdiffusion has also been identified in systems with long-range interactions [36–41] and short-range interacting systems subject to quasiperiodic potentials [42, 43], albeit with some controversies [44].

So far, the study of superdiffusion mainly focuses on closed systems since it is generally believed that coupling a system to the environment results in bulk dissipation, leading to diffusion. A well-studied example is the free fermion chain subject to local particle dephasing [45–52]. In this context, dephasing introduces finite lifetimes to the original free modes, resulting in a mean free path beyond which particle motion resembles a Gaussian random walk.

In contrast, our study identifies a superdiffusive transport in noninteracting fermion systems by generalizing the onsite dephasing to "quasi-particle" dephasing. The "quasi-particles" are defined as superpositions of fermions near position $x$:

$$\hat{d}_x = \sum_a d_a \hat{c}_{x+a}, \tag{1}$$

where the vector $d_a$ is assumed to be local near the origin. The momentum distribution characterizing these quasi-particles is

$$d_k \equiv \sum_a d_a e^{ika}. \tag{2}$$

Remarkably, our investigation unveils a direct link between the nodal structure of $d_k$ and the occurrence of superdiffusion:

1. When $d_k$ possesses a nodal point at generic momentum $k_o$ (with nonzero velocity $v_{k_o} \neq 0$) characterized by $|d_k| \sim (k - k_o)^n$, particle transport exhibits a ballistic front, and the dynamical scaling exponent is given by $z_n = (2n + 1)/(2n)$.

2. In cases where $d_k$ features a higher-order nodal point at zero-velocity point $k_o$, described as $|d_k| \sim (k - k_o)^n$ where $n \geq 2$, the particle transport exhibit a superdiffusive front, and the dynamical exponent is $z_n = (2n + 1)/(2n - 1)$.

We demonstrate the superdiffusion in dephasing models by analyzing the dynamics of the Wigner function [51,52]. This approach develops an effective description of transport behavior across extended temporal and spatial scales. The Wigner function framework translates the many-body transport problem into a single-particle random walk process. In this context, the presence of nodal points signifies the existence of long-lived modes with diverging mean free paths. The probabilistic distribution characterizing these mean free paths exhibits a heavy-tailed nature, a hallmark of the Lévy walk [53], a well-established model of superdiffusive processes.

Significantly, the dynamical exponent of the charge transport is intricately linked to the nodal structure of the dephasing quasi-particle. In instances where certain symmetries are present, the presence of nodal points becomes generic, leading to a robust manifestation of superdiffusion characterized by exact dynamical exponents. Besides, fine-tuning the nodal structure of quasi-particles enables the systematic generation of a spectrum of dynamical exponents. Our analytical approach extends naturally to higher-dimensional systems, underscoring the universality of our findings in the context of dephasing models.

## 2   Quasi-particle dephasing model

Consider the dynamics of a dephasing model governed by the Lindbladian:

$$\partial_t \hat{\rho} = -i[\hat{H}, \hat{\rho}] - \frac{\gamma}{2} \sum_x [\hat{L}_x, [\hat{L}_x, \hat{\rho}]], \tag{3}$$

where the Hamiltonian $\hat{H}$ represents a basic noninteracting fermion chain, given by

$$\hat{H} = \sum_i (\hat{c}_i^\dagger \hat{c}_{i+1} + \hat{c}_{i+1}^\dagger \hat{c}_i), \tag{4}$$

characterized by a group velocity $v_k = 2 \sin k$. The jump operator $\hat{L}_x = \hat{d}_x^\dagger \hat{d}_x$ captures the dephasing process affecting the quasi-particles $\hat{d}_x$. In the previous studies, particularly in the context of both monitored [51] and open systems [46], the case where $\hat{d}_x = \hat{c}_x$ has been well-explored, resulting in a clear demonstration of diffusive particle transport.

We first focus on the scenario involving quasi-particles with time-reversal and spatial-reflection symmetry. Specifically, we examine a scenario involving a three-site quasi-particle configuration:

$$\hat{d}_x = \frac{1}{\sqrt{2+a^2}} (\hat{c}_{x-1} - a\hat{c}_x + \hat{c}_{x+1}), \tag{5}$$

where $a$ is a real parameter. The corresponding momentum distribution is

$$d_k = \frac{2\cos k - a}{\sqrt{2+a^2}}. \tag{6}$$

In the range $-2 < a < 2$, $d_k$ exhibits two nodal points, $k_\pm = \pm \arccos(a/2)$, around which $d_k$ is linearly dispersed. Upon reaching $a = \pm 2$, these two nodal points merge into a higher-order nodal point at $k = 0$ or $k = \pi$ with quadratic dispersion: $d_k \propto \sin^2(k/2)$. For values $|a| > 2$, $d_k$ does not possess any nodal point.

Starting from a half-filling domain wall state

$$|\psi_o\rangle = |1\ldots 10\ldots 0\rangle, \tag{7}$$

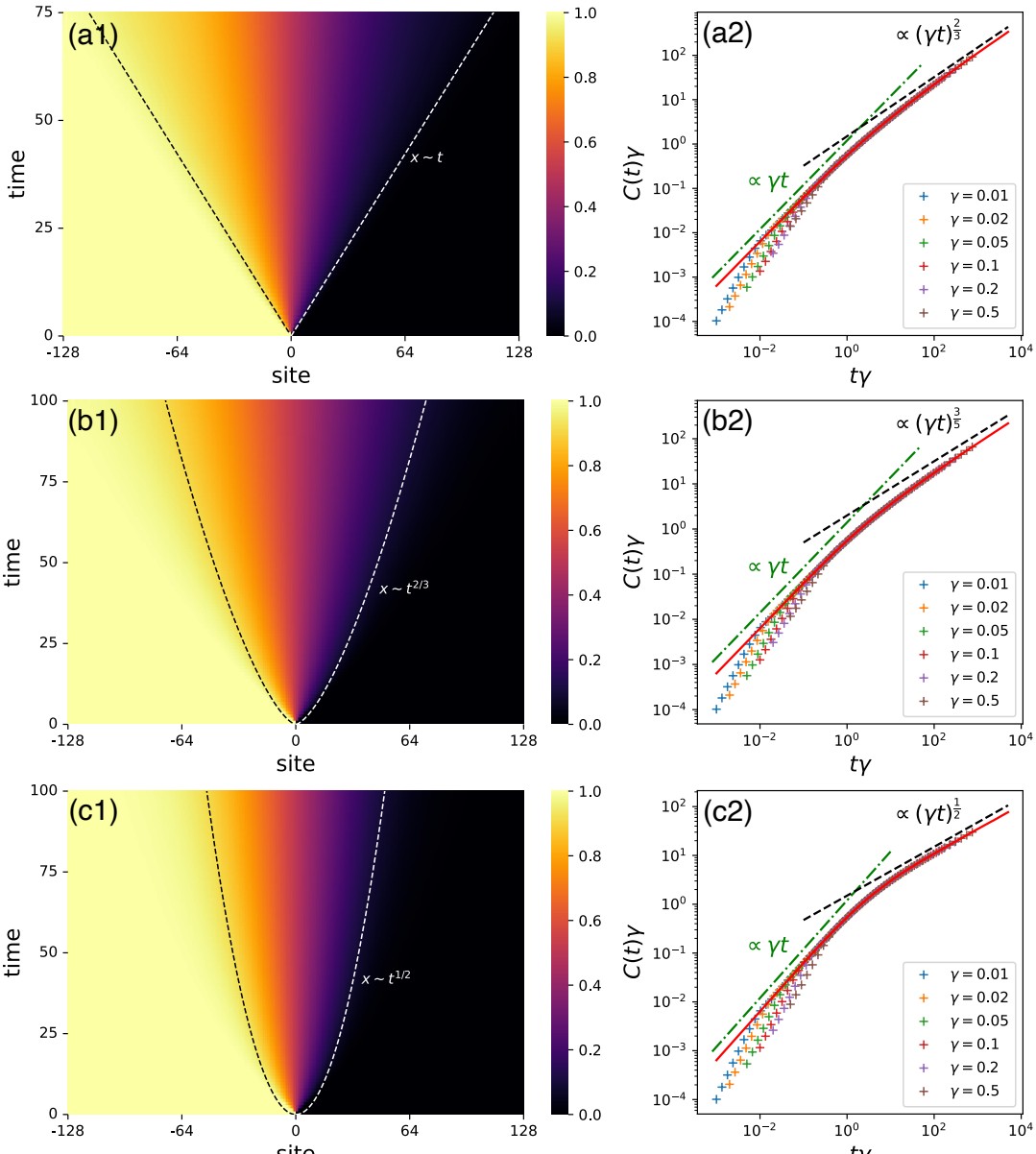

Figure 1: Numerical simulations of the particle transport of the dephasing Lindbladian Eq. (3), with $\hat{d}_x$ defined according to Eq. (5). Subplots are presented for different parameters, namely (a1)(a2) for $a = \sqrt{2}$, (b1)(b2) for $a = 2$, and (c1)(c2) for $a = 3$. Subplots (a1)(b1)(c1) display the density evolution of systems with different dephasing quasi-particles. The system sizes are fixed to $L = 256$, and the dephasing strength is $\gamma = 0.5$. The dynamics are initiated from the domain-wall state $|\psi_o\rangle = |1 \ldots 10 \ldots 0\rangle$, and feature either (a1) a ballistic wavefront, (b1) a superdiffusive wavefront, or (c1) a diffusive wavefront. Subplots (a2)(b2)(c2) shows the charge transport $C(t)$ for different dephasing quasi-particles. The system size for this simulation is fixed to $L = 3000$. For exact Lindbladian simulation, in the short time regime ($t < 1$), the transport behaviors deviate from the Wigner function dynamics results, following a $C(t) \propto t^2$ scaling. After $t > 1$, the Lindbiadian result approaches the Wigner function dynamics, which exhibits a crossover from ballistic ($z = 1$) to (a2) superdiffusive with dynamical exponent $z = 3/2$, (b2) superdiffusive with dynamical exponent $z = 5/3$, and (c2) diffusive with dynamical exponent $z = 2$.

we study the evolution of the particle density $\langle \hat{n}_i \rangle_t = \langle \hat{c}_i^\dagger \hat{c}_i \rangle_t$ as well as the transported charge $C(t) = \sum_{i \geq 1} \langle \hat{n}_i \rangle_t$ by computing the dynamics of the two-point correlation function $\langle \hat{c}_i^\dagger \hat{c}_j \rangle$ under Eq. (3). The Lindbladian with quadratic Hermitian jump operator $\hat{L}$ satisfies a closed hierarchy [54–56]. In this case, the evolution equation of $\langle \hat{c}_i^\dagger \hat{c}_j \rangle$ is closed (see Appendix A for deriving the correlation function dynamics). Therefore, we can numerically simulate the charge transport for system size up to $L = 3000$. For different choices of $a$, there are three distinct transport behaviors.

$a = \sqrt{2}$ **case**   From Fig. 1(a1), we see the density evolution features a ballistic front. The charge transport shows a scaling behavior (after $t > 1$):

$$\gamma C(t) \sim f(\gamma t), \tag{8}$$

wherein the scaling function exhibits asymptotic behavior:

$$f(x) \sim x^{2/3}, \quad \text{as} \quad x \to \infty.$$

This behavior indicates a dynamical exponent converging to

$$z = \frac{3}{2}, \tag{9}$$

in the long-time regime. Note that the scaling function does not conform to the KPZ universality class.

$a = 2$ **case**   As shown in Fig. 1(b1), the density evolution exhibits a superdiffusive front instead. After $t > 1$, the transport converges to the form [displayed in Fig. 1(b2)]

$$\gamma C(t) \sim g(\gamma t), \tag{10}$$

with a different scaling function

$$g(x) \sim x^{3/5},$$

in the large $x$ limit, indicating a dynamical exponent of

$$z = \frac{5}{3}. \tag{11}$$

$a = 3$ **case**   As demonstrated in Fig. 1(c1) [and in Fig. 1(c2) regarding the dynamical exponent], the transport displays apparent diffusive scaling in the long-time regime.

This observation underscores the close relationship between the nodal structure of the dephasing quasi-particle and the dynamical scaling of the transport.

## 3  Theoretical prediction of the dynamical exponent

### 3.1  Wigner dynamics

In Refs. [51, 52], the authors introduced a Wigner dynamics framework tailored for free fermion systems characterized by quadratic jump operators. The particle motion in the free fermion system is captured by the Wigner distribution [57]:

$$n(x, k, t) \equiv \sum_s e^{iks} \left\langle \hat{c}_{x+s/2}^\dagger \hat{c}_{x-s/2} \right\rangle_t. \tag{12}$$

This quantity essentially represents the particle density at position $x$ with momentum $k$ and offers a semiclassical perspective that accurately captures the system's dynamics in a coarse-grained sense. In Appendix B, we formally prove the exact Lindblad equation (3) leads to the following Wigner function dynamics:

$$\frac{\partial n}{\partial t}(x,k,t) = -2\sin k \frac{\partial n}{\partial x}(x,k,t) - \gamma|d_k|^2 n(x,k,t) + \gamma|d_k|^2 \int \frac{dq}{2\pi}|d_q|^2 n(x,q,t). \quad (13)$$

This equation describes a statistical process wherein a wave packet with momentum $k$ has a probability proportional to $|d_k|^2$ to shift to a different momentum. The probability distribution of the new momentum $q$ follows the distribution $|d_q|^2$. Note that the steady state solution to the equation of motion is $n(x,k) = $ const., which correspond to the original Lindblad equaiton Eq. (3) is unital and therefore admit the solution in each particle number sector $\mathcal{H}_N$ (the subspace spanned by $N$-particle states):

$$\rho_{\text{NESS}}^{(N)} = \frac{1}{\dim \mathcal{H}_N} \sum_{|\psi\rangle \in \mathcal{H}_N} |\psi\rangle\langle\psi|. \quad (14)$$

We proceed to solve this linear equation employing the Green's function method:

$$n(x,k,t) = (G * n_o)(x,k,t) = \int G(y,k,t)n_o(x-y,k)dy. \quad (15)$$

Taking the initial state as a domain wall configuration, i.e., $n_o(x,k) = \theta(-x)$, this expression simplifies to

$$n(x,k,t) = \int_x^\infty G(y,k,t)dy, \quad (16)$$

with the initial condition $G(x,k,0) = \delta(x)$. The Green's function $G(x,k,t)$ can be efficiently simulated via a random walk approach [51] involving the following steps:

1. The velocity is determined by momentum: $x'(t) = v[K(t)] = 2\sin K(t)$.

2. The quantity $K(t)$ remains constant within each interval $[t_0, t_1), [t_1, t_2), \cdots$, with each interval being independent and following an exponential distribution with an average value of $\overline{t_{i+1} - t_i} = \gamma^{-1}|d_k|^{-2}$.

3. The momenta $K_{i+1}$ are randomly distributed with a probability $p(k)$ proportional to $|d_k|^2$.

4. The probability density $p(x,k,t)$ corresponds to the Green's function $G(x,k,t)$, which can be determined numerically by sampling various random trajectories.

By employing this method and sampling multiple random trajectories, we obtain access to the scaling exponent in the long-time regime with high accuracy.

For the specific dephasing model involving quasi-particles in Eq. (5), Fig. 1(a2)(b2)(c2) showcase comparisons of charge transport between the exact Lindbladian dynamics on a 1D lattice and the Wigner dynamics. Initially distinct, the Lindblad dynamics gradually converges to the Wigner dynamics beyond $t > 1$. In Appendix C, we demonstrate an agreement in the dynamics of density profiles obtained through both methods, particularly evident in the long-time regime. This agreement supports the accuracy of the Wigner function description. Leveraging this validation, we extend our numerical simulation using Wigner dynamics, pushing the simulation time to $t > 10^7$. As shown in Fig. 2, this extension enables a precise showcase of the convergence of the dynamical scaling $\alpha(t)$ towards 2/3 and 3/5.

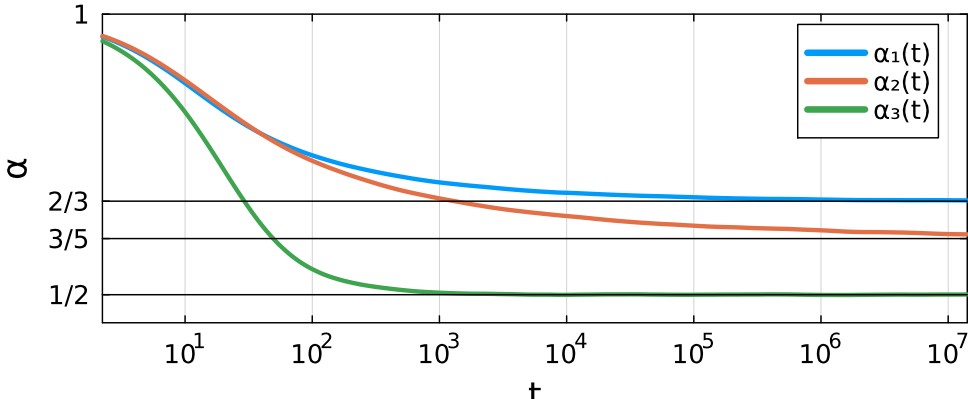

Figure 2: The dynamical exponent $\alpha(t) = d \log C(t)/d \log(t)$ of the charge transport for Wigner dynamics (13) with $\gamma = 0.1$. These results are obtained from the random walk simulations with $5 \times 10^6$ samples. The exponents $\alpha_1(t)$, $\alpha_2(t)$, and $\alpha_3(t)$ correspond to the cases $a = \sqrt{2}$, $a = 2$, and $a = 3$ respectively.

## 3.2 Lévy walk

In the random walk picture, we show that the nodal point in the momentum distribution leads to the phenomenon of Lévy walk [53]. In stark contrast to the Gaussian characteristics defining Brownian motion and standard diffusion, a Lévy walk constitutes a stochastic process dictated by a heavy-tailed probability distribution $p(l)$ governing step length $l$ of each transition. In the $l \to \infty$ limit, this distribution conforms to a power-law behavior:

$$p(l) \sim l^{-1-z}, \tag{17}$$

where $1 < z < 2$ is the Lévy exponent. When each step takes equal time, the cumulative displacement conforms to an asymptotic behavior:

$$X(t) = |\sum_i l_i| \sim t^{1/z}. \tag{18}$$

Hence, the dynamical exponent governing the system's behavior aligns with the Lévy exponent, confirming that a Lévy walk implies superdiffusion.

In systems where time-reversal and reflection symmetries are preserved, we define an indicator as

$$\nu = d_0 d_\pi, \tag{19}$$

which indicates a nontrivial condition when $\nu \leq 0$. For quasi-particles in Eq. (5), this corresponds to the range $-2 \leq a \leq 2$. A nontrivial $\nu < 0$ implies the existence of a nodal point at $k_o \in (0, \pi)$. We refer to this nodal point as a "generic nodal point." By introducing $q = |k - k_o|$, in the vicinity of $q \approx 0$, the mean free path exhibits the asymptotic behavior:

$$l_k \sim \tau_k \sim |d_k|^{-2} \sim q^{-2}. \tag{20}$$

As $q$ approaches zero, the mean free path diverges. A change of variable ($q \to l$) in the probabilistic distribution results in:

$$1 = \int p(q)dq \sim \int q^2 dq \sim \int l^{-1} d(l^{-1/2}) \sim \int l^{-5/2} dl,$$

leading to a free path distribution $p(l) \sim l^{-5/2}$. Since the average time step is constant:

$$\overline{t_{n+1} - t_n} = \int_k p(k)\tau_k = \frac{1}{\gamma},$$

this random walk behavior aligns with a Lévy walk, characterized by an exponent of $z = 3/2$, consistent with our numerical simulations in Fig. 1(a2).

When $\nu = 0$, $d_k$ possesses a nodal point at one of the high symmetry points, $k_o$, with a vanishing velocity

$$v_k \sim |k - k_o| = q.$$

The symmetry condition requires the dispersion of $d_k$ to be at least quadratic: $d_k \sim q^2$. As a result, the mean free path scales as

$$l_k \sim v_k \tau_k \sim q^{-3}.$$

Then a change of variable $(q \to l)$ leads to

$$1 = \int p(q)dq \sim \int q^4 dq \sim \int l^{-4/3} d\left(l^{-1/3}\right) \sim \int l^{-8/3}dl,$$

which yields $p(l) \sim l^{-8/3}$. The dynamical scaling exponent becomes $z = 5/3$, in accordance to Fig. 1(b2).

In cases where $\nu > 0$, there is typically no nodal point in $d_k$, resulting in a bounded mean free time: $\tau_k \leq \tau_{\max}$, and subsequently, a finite mean free path $l \leq l_{\max}$, resulting in ordinary diffusive behavior with $z = 2$, as shown in Fig. 1(c2).

# 4 Spectrum of exact dynamical critical exponents

## 4.1 Fine-tuning the dephasing quasi-particles

Beyond the symmetric setting, we can also leverage specific fine-tuned dephasing quasi-particles to attain higher-order dispersion near the nodal points. This diversity in dispersion yields various dynamical scaling behaviors.

Let us begin by considering a model with a single nodal point:

$$|d_k| \sim \sin^n\left[(k - k_o)/2\right].$$

This type of dispersion can be realized by selecting the following form for $\hat{d}_x$:

$$\hat{d}_x = \frac{1}{\sqrt{\mathcal{N}_n}} \sum_{a=0}^{n} e^{-iak_o}\hat{c}_{x+a},$$

where $\mathcal{N}_n$ is the normalization factor, with the explicit form

$$\mathcal{N}_n = \sum_{a=0}^{n} \binom{n}{a}^2 = \frac{4^n \Gamma(n + \frac{1}{2})}{\sqrt{\pi}\Gamma(n)}, \qquad \Gamma(x) \text{ is the Gamma function.}$$

In this case, $\hat{d}_x$ possesses a nodal point at $k_o$, exhibiting $n$-th order dispersion. Following similar derivations, the mean free path for momentum-$k$ wave packet is given by $l_k \sim q^{2n}$, and the distribution takes the form:

$$p(l) \sim l^{-1}\frac{d}{dl}(l^{-1/2n}) = l^{-1-(2n+1)/2n}.$$

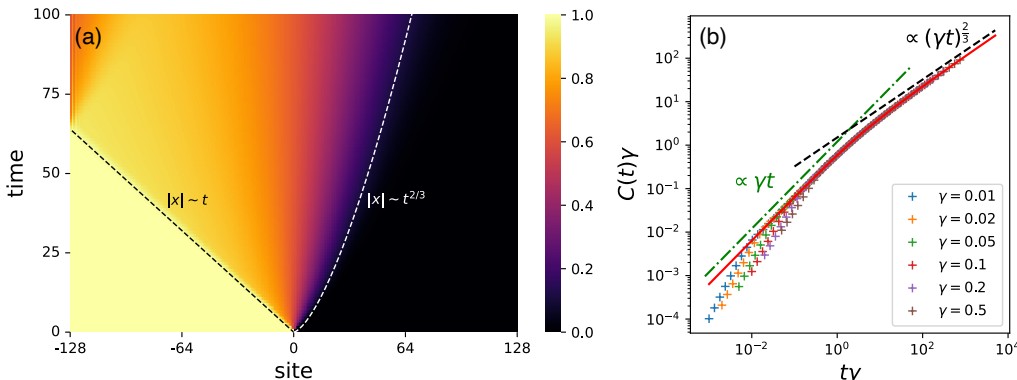

Figure 3: Numerical simulations of the (a) density evolution and the (b) particle transport of the quasi-particle dephasing system Eq. (3), with $\hat{d}_x$ defined according to Eq. (22). (a) the density evolution features a ballistic wavefront for the left part and a superdiffusive front for the right part. Due to the difference in wavefront velocities, by $t = 100$, the left front has already reached the boundary, whereas the right front has not. (b) charge transport shows superdiffusive scaling with $z = 3/2$.

The dynamical scaling exponents that can be tuned in this scenario are given by

$$z_n = \frac{2n+1}{2n}. \tag{21}$$

One simplest example is when $k_o = \pi/2$ and $n = 1$, in this case the dephasing quasiparticle is

$$\hat{d}_x = \frac{1}{\sqrt{2}}(\hat{c}_x - i\hat{c}_{x+1}). \tag{22}$$

Note that since the dephasing quasi-particle has no reflection symmetry, the left and right wavefronts are different, as displayed in Fig. 3(a). This behavior was previously observed in a monitored free fermion system [58], which shows a special skin effect in the steady state when adding certain feedback operations. The charge transport for this system shows superdiffusion with $z = 3/2$.

On the other hand, if we set $k_o = 0$ (or equivalently $k_o = \pi$), the velocity $v_k \sim k$, resulting in

$$l_k \sim v_k \tau_k \sim k^{-(2n-1)}.$$

Consequently, the probability distribution becomes:

$$p(l) \sim l^{-1-(2n+1)/(2n-1)},$$

which leads to the dynamical scaling exponent

$$z_n = \frac{2n+1}{2n-1}. \tag{23}$$

Note that the derivation is valid only for $n \geq 2$. In the $n = 1$ case, there would be usual diffusive transport.

## 4.2 Higher dimensional cases

The analysis extends to higher dimensions, where nodal structures can also be nodal lines and surfaces. To begin, the Wigner dynamics Eq. (13) naturally generalize to $D$-dimensions with

slight modifications:

$$\frac{\partial n}{\partial t}(\vec{x}, \vec{k}, t) = -2 \sum_{i=1}^{D} \sin(k_i) \frac{\partial n}{\partial x_i}(\vec{x}, \vec{k}, t) - \gamma |d_{\vec{k}}|^2 \left[ n(\vec{x}, \vec{k}, t) - \int \frac{d^D q}{(2\pi)^D} |d_{\vec{q}}|^2 n(\vec{x}, \vec{q}, t) \right]. \quad (24)$$

We refer to Appendix B for the proof. For simplicity, we assume the systems to be square/cubic lattices with nearest-neighbor hopping Hamiltonians, with dispersion relation

$$v_{\vec{k}} = 2(\sin k_1 \vec{e}_1 + \sin k_2 \vec{e}_2), \quad \text{and} \quad v_{\vec{k}} = 2(\sin k_1 \vec{e}_1 + \sin k_2 \vec{e}_2 + \sin k_3 \vec{e}_3)$$

respectively.

**2D systems** If the dephasing quasi-particle has the time-reversal and reflection symmetry, $d_{\vec{k}}$ is a real function on the Brillouin zone. We can then similarly define three independent indicators:

$$v_1 = d_{(0,0)} d_{(\pi,0)}, \qquad v_2 = d_{(0,0)} d_{(0,\pi)}, \qquad \text{and} \qquad v_3 = d_{(0,0)} d_{(\pi,\pi)}.$$

A negative value for these indicators indicates a nodal line $\vec{k}_o(\theta)$ in the Brillouin zone parametrized by $\theta \in [0, 1]$. In proximity to this nodal line, we expect consistent behavior with $b_{\vec{k}} \sim k_\perp^n$, where $k_\perp$ represents the local variable orthogonal to the $\vec{k}_o$ curve. The asymptotic probabilistic distribution $p(k_\perp)$ can be approximated by integrating out $k_\parallel$:

$$1 = \int p(\vec{k}) dk_\parallel dk_\perp \sim \int k_\perp^{2n} dk_\perp.$$

That is, $p(k_\perp) \sim k_\perp^{2n}$. The mean free path is then $l \sim k_\perp^{-2n}$, indicating a 2D Lévy walk with a dynamical scaling exponent given by

$$z_n = \frac{2n + 1}{2n}. \quad (25)$$

The nodal line intersects with a high-symmetry point if any indicator yields zero. The transport properties remain unchanged as they are determined by the segment of the curve with nonzero momentum.

When we relax the symmetry restriction on $\hat{d}_x$, $d_{\vec{k}}$ becomes a complex function on $k$. Even in this case, nodal lines can exist without fine-tuning. Consider the winding number

$$W[\mathcal{C}] = \oint_{\mathcal{C}} \frac{d_{\vec{k}}}{|d_{\vec{k}}|}, \quad (26)$$

of a contractible loop $\mathcal{C}$ in the Brillouin zone. A nonzero winding number signifies the presence of a nodal point $k_o$ within the loop. For this analysis, we assume

$$p(\vec{k}) \sim |\vec{k} - \vec{k}_o|^{2n} \equiv |\vec{q}|^{2n},$$

in the vicinity of the nodal point. For a generic nodal point $k_o \neq 0$, the mean free path is given by

$$l_{\vec{k}} \sim \tau_{\vec{k}} \sim |\vec{q}|^{-2n},$$

and the probability distribution becomes:

$$1 = \int p(\vec{q}) d\vec{q}^2 \sim \int q^{2n} q \, dq \sim \int l^{-2-1/n} dl.$$

Consequently, the dynamical exponent is

$$z_n = \frac{n+1}{n}\,. \tag{27}$$

If the nodal point is situated at one of the high-symmetry points, the mean free path becomes

$$l_{\vec{k}} \sim v_{\vec{k}} \tau_{\vec{k}} \sim |\vec{q}|^{-(2n-1)}\,,$$

and the probability distribution can be expressed as

$$1 = \int p(\vec{q})dq^2 \sim \int l^{-2-3/(2n-1)}dl\,.$$

The dynamical exponent in this case is

$$z_n = \frac{2n+2}{2n-1}\,. \tag{28}$$

For superdiffusion in this context, $n \geq 3$ is required; a smaller value of $n$ results in diffusive transport.

**3D systems**  If time-reversal and reflection symmetry are present, we can similarly define seven indicators, which are the product of $\vec{d}_0$ and $\vec{d}_{\vec{k}}$ at one of seven high-symmetry points in the Brillouin zone. A negative sign in these indicators implies the presence of a nodal surface. Assuming that the dispersion near the nodal surface is proportional to the orthogonal component: $d_{\vec{k}} \sim k_{\perp}^n$, a similar calculation yields a dynamical exponent of

$$z_n = \frac{2n+1}{2n}\,. \tag{29}$$

Without the symmetry constraint, we can similarly define winding number $W[\mathcal{C}]$ for a contractible loop $\mathcal{C}$; a non-zero winding implies a nodal line. Assuming the dispersion relation $d_{\vec{k}} \sim k_{\perp}^n$ near the curve, we obtain a dynamical exponent of

$$z_n = \frac{n+1}{n}\,. \tag{30}$$

## 5 Conclusion

This study uncovered a straightforward yet profound mechanism that leads to superdiffusive transport within noninteracting fermion systems subjected to local dephasing. Our findings demonstrate that we can fundamentally alter the system's behavior by extending the onsite particle dephasing to the dephasing of local quasi-particles featuring nodal points. The dynamics of a momentum-$\vec{k}$ wave packet in this setting resemble the diffusive particle but with a unique feature: its mean free paths $l_{\vec{k}}$ diverge when the momentum approaches the nodal point:

$$l_{\vec{k}} \sim \left|\vec{k} - \vec{k}_o\right|^n\,. \tag{31}$$

By studying the Wigner dynamics of the Lindbladian, we have rigorously mapped the system's behavior to that of a random walk. Notably, this random walk manifests as a *Lévy walk*, a well-established model of superdiffusion in physics. This mapping not only elucidates the physical underpinnings of the observed superdiffusion but also enables us to determine the dynamical exponent governing the system's behavior precisely.

Furthermore, it empowers us to design and engineer models with different exact dynamical exponents, broadening our grasp of the phenomenon. It is worth noting that this superdiffusive transport extends naturally to higher dimensions. This generality underscores the universality of the mechanism, offering valuable insights that can be applied across a spectrum of quantum many-body systems.

In this work, the initial state of the dynamics is consistently set to the domain wall state

$$|\psi_o\rangle = |1\cdots 10\cdots 0\rangle\,.$$

However, the phenomenon of superdiffusion is not confined to this specific initial state. According to the random walk argument, superdiffusive charge transport should also be present in a generic, imbalanced configuration and in the infinite temperature ensemble. Notably, further research [59] demonstrates that the model presented in this paper exhibits superdiffusion, with the same dynamical exponent, in a boundary driving setup, which is considered indicative of infinite-temperature transport [1,60].

Additionally, the concept of nodal points in quasi-particles provides a general strategy for achieving superdiffusion. This concept also leeds to new anomalous transport phenomena in disordered systems [61,62]. In Refs. [63,64], it is demonstrated that nodal impurities lead to new types of superdiffusive behaviors.

## Acknowledgments

J.R. and Y.-P. W. thanks Marko Žnidarič for his careful reading of the manuscript and useful comments and suggestions. The numerical simulation of the Lindblad equation uses the `Julia` package `DifferentialEquation.jl` [65].

**Funding information** This work is supported by the National Natural Science Foundation of China (NSFC) under grant number 12325404, the Chinese Academy of Sciences under grant number XDB33020000, and the National Key R&D Program of China under grant numbers 2022YFA1403800 and 2023YFA1406704. J.R. acknowledges support from the Leverhulme Trust Research Leadership Award RL-2019-015.

## A    Closed hierarchy of the correlation function

In this section, we will show that the dynamics governed by a Lindbladian consisting of free fermion Hamiltonian and Hermitian quadratic jump operators can be efficiently simulated due to a closed hierarchy [54–56] of the correlation function. Specifically, the dynamics of the two-point correlation function can be formulated as a differential equation that is linear in itself and does not involve any multi-point correlations.

We first consider the Lindblad equation for operators:

$$\partial_t \hat{O} = i[\hat{H}, \hat{O}] - \frac{\gamma}{2}\sum_n [\hat{L}_n[\hat{L}_n, \hat{O}]]\,, \tag{A.1}$$

where each jump operator is a Hermitian fermion bilinear:

$$\hat{L}_x = \sum_{ab} d_a^* d_b \hat{c}_{x+a}^\dagger \hat{c}_{x+b} \equiv \sum_{ij} A_{x,ij}\hat{c}_i^\dagger \hat{c}_j\,, \qquad A_{x,ij} = d_{i-x}^* d_{j-x}\,. \tag{A.2}$$

Since we concern only the two-point correlation $G_{ij} = \langle c_i^\dagger c_j \rangle$, we can choose $\hat{O}_{ij} = c_i^\dagger c_j$. Using the commutation relation $[c_i^\dagger c_j, c_k^\dagger c_l] = \delta_{jk} c_i^\dagger c_l - \delta_{il} c_k^\dagger c_j$, we know the following identity:

$$\sum_{kl} [A_{kl} \hat{c}_k^\dagger \hat{c}_l, \hat{c}_i^\dagger \hat{c}_j] = \sum_k [A_{ki} \hat{c}_k^\dagger \hat{c}_j - \hat{c}_i^\dagger \hat{c}_k A_{jk}]. \tag{A.3}$$

We can use the identity to calculate the commutator of two fermion bilinears and obtain the following:

$$i \sum_{kl} H_{kl} [\hat{c}_k^\dagger \hat{c}_l, \hat{O}_{ij}] = i \sum_{kl} H_{kl} (\delta_{il} \hat{c}_k^\dagger \hat{c}_j - \delta_{jk} \hat{c}_i^\dagger \hat{c}_l) = i[H^T \cdot \hat{O} - \hat{O} \cdot H^T]_{ij}.$$

Similarly, the double commutation in the second term is:

$$-\frac{\gamma}{2} \sum_x [\hat{L}_x [\hat{L}_x, \hat{O}_{ij}]] = -\frac{\gamma}{2} \sum_x [(A_x^*)^2 \cdot \hat{O} + \hat{O} \cdot (A_x^*)^2 - 2A_x^* \cdot \hat{O} \cdot A_x^*].$$

Together, the EOM of the two-point correlation function is

$$\partial_t G = X^\dagger \cdot G + G \cdot X + \gamma \sum_x A_x^* \cdot G \cdot A_x^* \equiv \mathcal{L}[G], \tag{A.4}$$

where $X = -iH^* - \frac{\gamma}{2} \sum_x (A_x^*)^2$.

Note that the right-hand side of Eq. (A.4) is linear in $G$, the evolution of $G$ can be formally written as

$$G(t) = e^{\mathcal{L}t} [G_0]. \tag{A.5}$$

To obtain the trajectory in the numerical simulation, simply implement the $\mathcal{L}[\cdot]$ action and insert the linear operator into a numerical solver for the differential equation.

# B  Wigner function dynamics

In this appendix, following Refs. [51,52], we derive the Wigner dynamics of quasi-free Lindbiadian for general $d$-dimension. The Hamiltonians are supposed to be the simplest free fermion model on the square lattice:

$$\hat{H} = \sum_{\langle \vec{x}, \vec{y} \rangle} \hat{c}_{\vec{x}}^\dagger \hat{c}_{\vec{y}} + \hat{c}_{\vec{y}}^\dagger \hat{c}_{\vec{x}}. \tag{B.1}$$

The Lindblad equation for operator $\hat{O}$ has the form:

$$\partial_t \hat{O} = i[\hat{H}, \hat{O}] - \frac{\gamma}{2} \sum_n [\hat{L}_n [\hat{L}_n, \hat{O}]]. \tag{B.2}$$

We are considering the evolution of the operator

$$\hat{n}(\vec{x}, \vec{k}) \equiv \sum_s e^{i\vec{k} \cdot \vec{s}} \hat{c}_{\vec{x}+\frac{\vec{s}}{2}}^\dagger \hat{c}_{\vec{x}-\frac{\vec{s}}{2}}. \tag{B.3}$$

The Wigner function is then obtained by taking the expectation value: $n(\vec{x}, \vec{k}, t) = \langle \hat{n}(\vec{x}, \vec{k}) \rangle_t$.

**Hopping Hamiltonian**   We first consider the Hamiltonian part of the Lindbladian. Using the identity

$$\left[\hat{c}_i^\dagger\hat{c}_j,\hat{c}_k^\dagger\hat{c}_l\right]=\hat{c}_i^\dagger[\hat{c}_j,\hat{c}_k^\dagger\hat{c}_l]+[\hat{c}_i^\dagger,\hat{c}_k^\dagger\hat{c}_l]\hat{c}_j=\delta_{jk}\hat{c}_i^\dagger\hat{c}_l-\delta_{il}\hat{c}_k^\dagger\hat{c}_j\,,\tag{B.4}$$

we obtain the commutation relation

$$[\hat{H},\hat{c}_{\vec{x}}^\dagger\hat{c}_{\vec{y}}]=\sum_{i=1}^{D}\left(\hat{c}_{\vec{x}+\vec{e}_i}^\dagger\hat{c}_{\vec{y}}+\hat{c}_{\vec{x}-\vec{e}_i}^\dagger\hat{c}_{\vec{y}}-\hat{c}_{\vec{x}}^\dagger\hat{c}_{\vec{y}+\vec{e}_i}-\hat{c}_{\vec{x}}^\dagger\hat{c}_{\vec{y}-\vec{e}_i}\right)\,,\tag{B.5}$$

where $\vec{e}_i$ is the unit vector for each direction and thus

$$i\left[\hat{H},\hat{n}(\vec{x},\vec{k})\right]=i\sum_{\vec{s}}e^{i\vec{k}\cdot\vec{s}}\sum_{i=1}^{D}\left[\hat{c}_{\vec{x}+\frac{\vec{s}}{2}+\vec{e}_i}^\dagger\hat{c}_{\vec{x}-\frac{\vec{s}}{2}}+\hat{c}_{\vec{x}+\frac{\vec{s}}{2}-\vec{e}_i}^\dagger\hat{c}_{\vec{x}-\frac{\vec{s}}{2}}-\hat{c}_{\vec{x}+\frac{\vec{s}}{2}}^\dagger\hat{c}_{\vec{x}-\frac{\vec{s}}{2}+\vec{e}_i}-\hat{c}_{\vec{x}+\frac{\vec{s}}{2}}^\dagger\hat{c}_{\vec{x}-\frac{\vec{s}}{2}-\vec{e}_i}\right]$$

$$=2\sum_{i=1}^{D}\sin(k_i)\left[\hat{n}\left(\vec{x}+\frac{\vec{e}_i}{2},\vec{k}\right)-\hat{n}\left(\vec{x}-\frac{\vec{e}_i}{2},\vec{k}\right)\right]\,.$$

In the coarse-grained

$$n\left(\vec{x}+\frac{\vec{e}_i}{2},\vec{k},t\right)-n\left(\vec{x}-\frac{\vec{e}_i}{2},\vec{k},t\right)\simeq\frac{\partial n}{\partial x_i}(\vec{x},\vec{k},t)\,.\tag{B.6}$$

So the Hamiltonian part of the dynamics is

$$dn(\vec{x},\vec{k},t)=-2\sum_{i=1}^{D}\sin(k_i)\frac{\partial n}{\partial x_i}(\vec{x},\vec{k},t)dt\,.\tag{B.7}$$

**Dissipation**   Here, we consider the dephasing of the local quasi-particle at $y$,

$$\hat{L}_y=\sum_{ab}d_a^*d_b\hat{c}_{y+a}^\dagger\hat{c}_{y+b}\,.\tag{B.8}$$

We denote $A_{ab}=d_a^*d_b$, the commutator $[\hat{L}_y,\hat{n}(x,k)]$ is

$$\left[\hat{L}_y,\hat{n}(x,k)\right]=\sum_{s,ab}e^{iks}A_{ab}\left[\hat{c}_{y+a}^\dagger\hat{c}_{y+b},\hat{c}_{x+\frac{s}{2}}^\dagger\hat{c}_{x-\frac{s}{2}}\right]$$

$$=\sum_{s,ab}e^{iks}A_{ab}\left[\delta_{y-x+b,\frac{s}{2}}\hat{c}_{y+a}^\dagger\hat{c}_{x-\frac{s}{2}}-\delta_{x-y-a,\frac{s}{2}}\hat{c}_{x+\frac{s}{2}}^\dagger\hat{c}_{y+b}\right]$$

$$=\sum_{ab}A_{ab}\left[e^{-2ik(x-y-b)}\hat{c}_{y+a}^\dagger\hat{c}_{2x-y-b}-e^{+2ik(x-y-a)}\hat{c}_{2x-y-a}^\dagger\hat{c}_{y+b}\right]\,.$$

Using the fact

$$\frac{1}{N}\sum_p e^{-ipa}\hat{n}(x,p)=\frac{1}{N}\sum_p\sum_s e^{ip(s-a)}\hat{c}_{x+\frac{s}{2}}^\dagger\hat{c}_{x-\frac{s}{2}}=\sum_p\delta_{s,a}\hat{c}_{x+\frac{s}{2}}^\dagger\hat{c}_{x-\frac{s}{2}}=\hat{c}_{x+\frac{a}{2}}^\dagger\hat{c}_{x-\frac{a}{2}}\,,$$

the result is

$$\left[\hat{L}_y,\hat{n}(x,k)\right]=\frac{1}{N}\sum_{ab,p}A_{ab}e^{-ip(a-b)}$$

$$\times\left[e^{-2i(k-p)(x-y-b)}\hat{n}\left(x+\frac{a-b}{2},p\right)-e^{+2i(k-p)(x-y-a)}\hat{n}\left(x-\frac{a-b}{2},p\right)\right]\,.$$

For the double commutator $\left[\hat{L}_y,\left[\hat{L}_y,\hat{n}(x,k)\right]\right]$, we need to replace the Wigner distribution in the right-hand side with $\left[\hat{L}_y,\hat{n}\right]$. There are four terms involved:

$$\left[\hat{L}_y,\left[\hat{L}_y,\hat{n}(x,k)\right]\right]=S_1+S_2+S_3+S_4,$$

where $S_1$ and $S_2$ come from

$$\hat{n}\left(x+\frac{a-b}{2},p\right)\longrightarrow\left[\hat{L}_y,\hat{n}\left(x+\frac{a-b}{2},p\right)\right];$$

the $S_3$ and $S_4$ come from

$$\hat{n}\left(x-\frac{a-b}{2},p\right)\longrightarrow\left[\hat{L}_y,\hat{n}\left(x-\frac{a-b}{2},p\right)\right].$$

In the following, we will simplify the expression term by term. For the first term $S_1$,

$$\begin{aligned}
S_1 &= \frac{1}{N^2}\sum_{abcd,pq,y}A_{ab}A_{cd}e^{-ip(a-b)-iq(c-d)}e^{-2i(k-p)(x-y-b)}e^{-2i(p-q)(x+\frac{a-b}{2}-y-d)}\hat{n}\left(x+\frac{a-b+c-d}{2},q\right)\\
&= \frac{1}{N}\sum_{abcd,pq}\left(\sum_y\frac{e^{2iy(k-q)}}{N}\right)A_{ab}A_{cd}e^{-ip(a-b)-iq(c-d)-2i(k-p)(x-b)}e^{-2i(p-q)(x+\frac{a-b}{2}-d)}\hat{n}\left(x+\frac{a-b+c-d}{2},q\right)\\
&= \frac{1}{N}\sum_{abcd,pq}\delta_{k,q}A_{ab}A_{cd}e^{-ip(a-b)-iq(c-d)-2i(k-p)(x-b)}e^{-2i(p-q)(x+\frac{a-b}{2}-d)}\hat{n}\left(x+\frac{a-b+c-d}{2},q\right)\\
&= \frac{1}{N}\sum_{abcd}\sum_p A_{ab}A_{cd}e^{-ip(a-b)-ik(c-d)}e^{2i(k-p)(\frac{a+b}{2}-d)}\hat{n}\left(x+\frac{a-b+c-d}{2},k\right)\\
&= \sum_{abcd}\left(\frac{1}{N}\sum_p e^{-2ip(a-d)}\right)A_{ab}A_{cd}e^{ik(a+b-c-d)}\hat{n}\left(x+\frac{a-b+c-d}{2},k\right)\\
&= \sum_{bc}(A^2)_{cb}e^{ik(b-c)}\hat{n}\left(x-\frac{b-c}{2},k\right).
\end{aligned}$$

The calculation for $S_4$ is similar to $S_1$:

$$\begin{aligned}
S_4 &= \frac{1}{N^2}\sum_{abcd,pq,y}A_{ab}A_{cd}e^{-ip(a-b)-iq(c-d)}e^{2i(k-p)(x-y-a)}e^{2i(p-q)(x-\frac{a-b}{2}-y-c)}\hat{n}\left(x-\frac{a-b+c-d}{2},q\right)\\
&= \frac{1}{N}\sum_{abcd,pq}\delta_{k,q}A_{ab}A_{cd}e^{-ip(a-b)-iq(c-d)}e^{2i(k-p)(x-a)}e^{2i(p-q)(x-\frac{a-b}{2}-c)}\hat{n}\left(x-\frac{a-b+c-d}{2},q\right)\\
&= \frac{1}{N}\sum_{abcd,p}A_{ab}A_{cd}e^{-ip(a-b)-ik(c-d)}e^{i(p-k)(a+b-2c)}\hat{n}\left(x-\frac{a-b+c-d}{2},k\right)\\
&= \sum_{abcd}\delta_{b,c}A_{ab}A_{cd}e^{-ik(a+b-c-d)}\hat{n}\left(x-\frac{a-b+c-d}{2},k\right)\\
&= \sum_{ad}(A^2)_{ad}e^{-ik(a-d)}\hat{n}\left(x-\frac{a-d}{2},k\right).
\end{aligned}$$

Since $\hat{L}_y$ is a particle number operator, $\hat{L}_y^2=\hat{L}_y$, i.e., $A^2=A$. Moreover, in the coarse-grained limit, we can approximate $\hat{n}(x-(b-c)/2,k)$ and $\hat{n}(x-(a-d)/2,k)$ with $\hat{n}(x,k)$. Therefore,

$$S_1=S_4=\sum_a d_a^* e^{-ika}\sum_b d_b e^{ikb}\hat{n}(x,k)=|d_k|^2\hat{n}(x,k).$$

Now we consider the $S_2$ part:

$$S_2 = -\frac{1}{N^2}\sum_{abcd,pq,y}A_{ab}A_{cd}e^{-ip(a-b)-iq(c-d)}e^{-2i(k-p)(x-y-b)}e^{2i(p-q)(x+\frac{a-b}{2}-y-c)}\hat{n}\left(x+\frac{a-b-c+d}{2},q\right)$$

$$= -\frac{1}{N}\sum_{abcd,pq}\left(\sum_y\frac{e^{2iy(k-2p+q)}}{N}\right)A_{ab}A_{cd}e^{-ip(a-b)-iq(c-d)-2i(k-p)(x-b)}e^{2i(p-q)(x+\frac{a-b}{2}-c)}\hat{n}\left(x+\frac{a-b-c+d}{2},q\right)$$

$$= -\frac{1}{N}\sum_{abcd,p}A_{ab}A_{cd}e^{-ip(a-b)-i(2p-k)(c-d)}e^{i(k-p)(a+b-2c)}\hat{n}\left(x+\frac{a-b-c+d}{2},2p-k\right)$$

$$= -\sum_{abcd}\left(\sum_p\frac{e^{-2ip(a-d)}}{N}\right)A_{ab}A_{cd}e^{ik(a+b-c-d)}\hat{n}\left(x+\frac{a-b-c+d}{2},2p-k\right)$$

$$= -\sum_{abcd}A_{ab}A_{cd}e^{ik(b-c)}\frac{1}{N}\sum_q e^{-iq(a-d)}\hat{n}\left(x+\frac{a-b-c+d}{2},q\right).$$

Using the coarse-graining approximation and replacing the momentum sum with the integral, we have:

$$S_2 \simeq -\sum_{bc}d_b d_c^* e^{ik(b-c)}\int\frac{d^D q}{(2\pi)^D}\sum_{ad}d_a^* d_d e^{-iq(a-d)}\hat{n}(x,q) = -|d_k|^2\int\frac{d^D q}{(2\pi)^D}|d_q|^2\hat{n}(x,q).$$

Straightforward calculation shows $S_3 \simeq S_2$:

$$S_3 = -\frac{1}{N^2}\sum_{abcd,pq,y}A_{ab}A_{cd}e^{-ip(a-b)-iq(c-d)}e^{2i(k-p)(x-y-a)}e^{-2i(p-q)(x+\frac{a-b}{2}-y-d)}\hat{n}\left(x-\frac{a-b-c+d}{2},q\right)$$

$$= -\frac{1}{N}\sum_{abcd,pq}\delta_{q,2p-k}A_{ab}A_{cd}e^{-ip(a-b)-iq(c-d)}e^{2i(k-p)(x-a)}e^{-2i(p-q)(x-\frac{a-b}{2}-d)}\hat{n}\left(x-\frac{a-b-c+d}{2},q\right)$$

$$= -\sum_{abcd}A_{ab}A_{cd}e^{-ik(a+b-c-d)}\int\frac{d^D q}{(2\pi)^D}e^{2ip(b-c)}\hat{n}\left(x-\frac{a-b-c+d}{2},2p-k\right)$$

$$= -\sum_{abcd}A_{ab}A_{cd}e^{-ik(a-d)}\int\frac{d^D q}{(2\pi)^D}e^{iq(b-c)}\hat{n}\left(x-\frac{a-b-c+d}{2},q\right)$$

$$\simeq -|d_k|^2\int\frac{d^D q}{(2\pi)^D}|d_q|^2\hat{n}(x,q).$$

Therefore, we have proved that the dynamics of the Wigner distribution is

$$\partial_t n(\vec{x},\vec{k}) = -2\sum_{i=1}^D\sin(k_i)\partial_{x_i}n(\vec{x},\vec{k})-\gamma|d_{\vec{k}}|^2 n(\vec{x},\vec{k})+\gamma|d_{\vec{k}}|^2\int\frac{d^D q}{(2\pi)^D}|d_{\vec{q}}|^2 n(\vec{x},\vec{q}). \quad (B.9)$$

## C Comparison between exact Lindblad and Wigner dynamics

In this appendix, we compare the exact Lindblad dynamics

$$\frac{\partial\rho}{\partial t} = -i[\hat{H},\rho]-\frac{\gamma}{2}\sum_x\left[\hat{d}_x^\dagger\hat{d}_x,[\hat{d}_x^\dagger\hat{d}_x,\rho]\right], \quad (C.1)$$

with the Wigner dynamics Eq. (B.9).

In terms of the dynamics of the density, we first notice that at short times, as shown in Fig. 4 for quasiparticle

$$\hat{d}_x = \frac{1}{2}\left(\hat{c}_{x-1}-\sqrt{2}\hat{c}_x+\hat{c}_{x+1}\right). \quad (C.2)$$

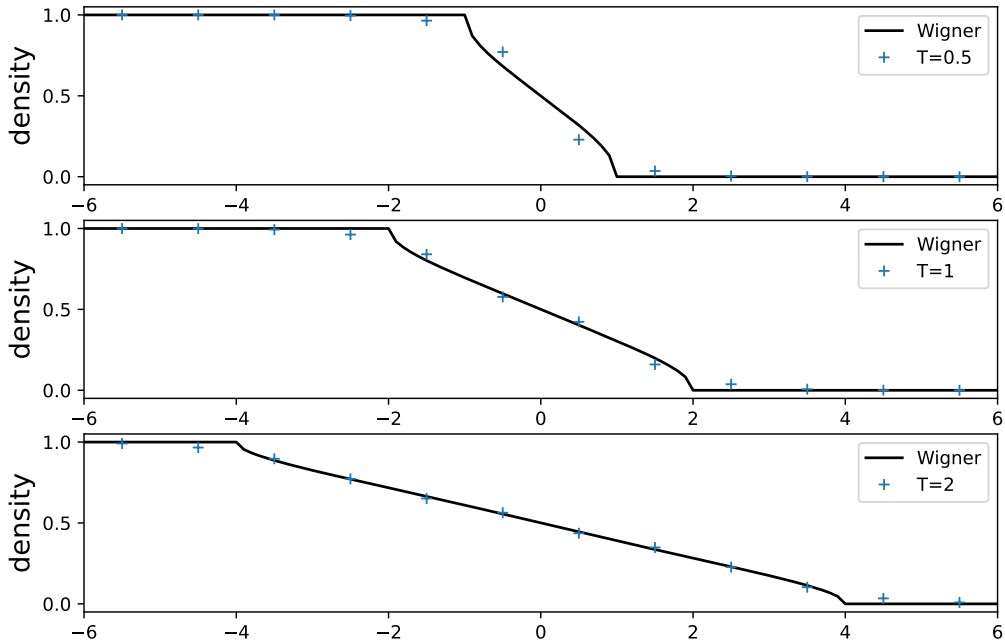

Figure 4: Comparison of the Lindblad equation (C.1) and the Wigner dynamics (B.9) at $T = 0.5, 1, 2$, with $a = \sqrt{2}$ and $\gamma = 0.5$. The markers show the results from the Lindblad equation, and the solid line represents the results of Wigner dynamics.

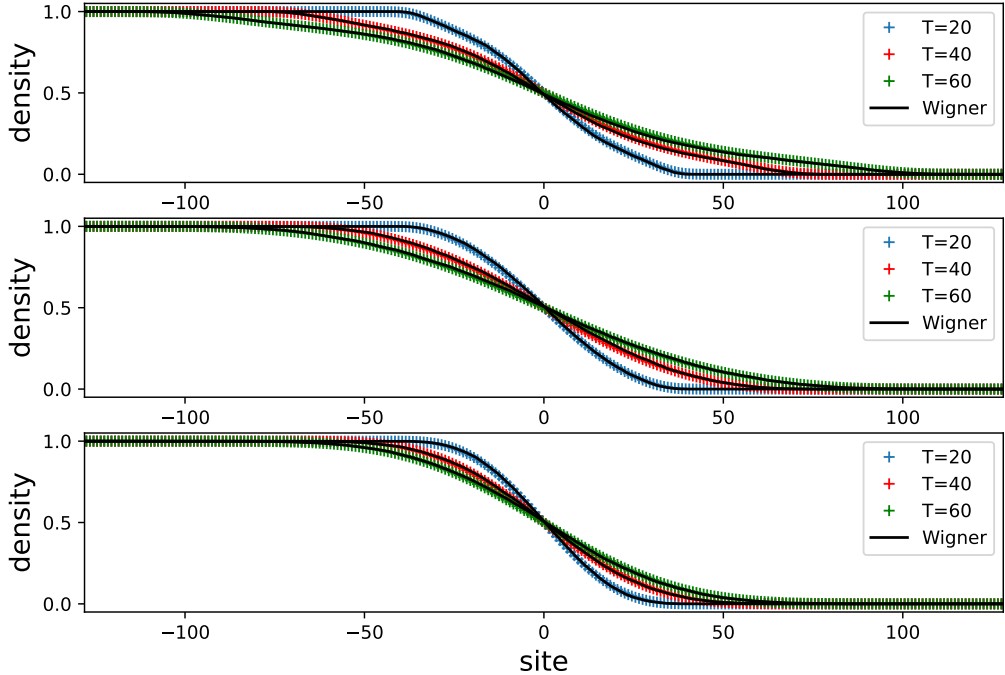

Figure 5: Comparison of the Lindblad equation (C.1) and the Wigner dynamics (B.9) at $T = 20, 40, 60$, with $a = \sqrt{2}$ (top), $a = 2$ (middel), $a = 3$ (bottom), and $\gamma = 0.5$. The markers show the results from the Lindblad equation, and the solid line represents the results of Wigner dynamics.

There are certain disagreements between the Lindblad dynamics and the Wigner dynamics. However, as time grows, the disagreements become less prominent.

The comparison at later times ($t = 20, 40, 60$), as displayed in Fig. 5 for quasiparticles

$$\hat{d}_x = \frac{1}{\sqrt{2 + a^2}} \left( \hat{c}_{x-1} - a\hat{c}_x + \hat{c}_{x+1} \right), \quad a = \sqrt{2}, 2, 3, \tag{C.3}$$

discussed in the main text shows good agreement.

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
