# Peer review of "Superdiffusive transport in quasi-particle dephasing models"

_SciPost Physics, doi:SciPost Phys. 17, 150 (2024)_

## Round 1 · Referee Report · Anonymous (Referee 1) · 2024-9-6

Report

The paper discusses superdiffusive spreading of the wavefront of fermions in a non-interacting model with "dispersive" jump operators in the Lindbladian. It is a cute and interesting idea and could be published in SciPost Physics. Our comments and suggestions are below:

1) On page 5, the authors seemed to imply that a difference with KPZ universality is the presence of a propagating wavefront? That claim would be false. The KPZ class arises as the endpoint of the instability of the 1+1d Navier-Stokes equations, for example -- the hydrodynamic modes here are propagating sound waves but with superdiffusive spreading. It's likely that this system does not have KPZ universality even if z=3/2, but the presence of the wavefront is not the reason for the difference. We suggest editing the wording here and throughout the paper to ensure that the reader is not misled about this point.

2) On the use of the term "hydrodynamics" -- hydrodynamics is an emergent description of chaotic many-particle systems, and would not apply to a free fermion model. The paper seems to the phrase "hydrodynamics" in awkward ways and we suggest avoiding this terminology. The paper has an interesting model but it may not be related at all to hydrodynamic phenomena more generally, and the robustness of the phenomena described here to interactions is far from clear, and we would be surprised if the scaling exponents were robust.

3) In Eq.(13) and Appendix B, the authors write down a kind of "master equation" for d_t n(x,k). This equation seems to admit the homogeneous solution n(x,k) = constant, which we found very surprising at first. If this is actually to be expected because the Lindblad jump operators are dephasing and don't correspond to single particle loss, some clarity would be appreciated; some comments in the main text would be helpful.

4) It would be ideal if Eq. (13) could be used to derive the dynamical critical exponent, rather than relying on a heuristic probabilistic argument. This seems doable to us, especially given the previous point: the modes which do not relax to some featureless local equilibrium are probably the ones near the point k_o, so modes in n(x,k) near k_o should be largely responsible for the effect.

5) Further comments on how local the dephasing jump operators need to be would be helpful, given that there is structure in Fourier space.

Recommendation

Ask for minor revision

  • validity: -
  • significance: -
  • originality: -
  • clarity: -
  • formatting: -
  • grammar: -

Author:  Jie Ren  on 2024-10-07  [id 4845]

(in reply to Report 3 on 2024-09-06)

We thank the reviewer for their careful reading and generally positive evaluation of our manuscript. Below are our responses to the comments and suggestions:

1) We appreciate the reviewer’s insight regarding the propagation of wavefronts in KPZ universality. We recognize that the presence of a propagating wavefront is not inherently a distinction from KPZ universality. In the revised manuscript, we will clarify that the scaling function in our system differs from KPZ, and we attribute this to differing physical origins without making unsupported claims. We will ensure that the text does not mislead readers about this point. 2) We adopted the term "hydrodynamics" from literature that discusses the dynamics of the Wigner function under the framework of "generalized hydrodynamics." We agree with the reviewer that using "hydrodynamics" in our context might be inappropriate for a free fermion model. We will replace "hydrodynamics" with "Wigner function dynamics" throughout the revised manuscript to avoid any confusion. 3) We agree with the reviewer’s observation regarding the homogeneous solution. This can indeed be attributed to the fact that the original Lindbladian is unital, implying that the nonequilibrium steady state is the maximally mixed state (subject to a fixed particle number). The local particle density in this maximally mixed state should be homogeneous, and the correct coarse-grained dynamics should reflect this feature. We have added comments in the main text to clarify this point as suggested. 4) We acknowledge the reviewer’s suggestion to derive the dynamical critical exponent directly from Eq. (13). While we agree that this is a minimalistic approach, our current method using a heuristic probabilistic argument follows directly from existing literature and provides an exact simulation framework for the Wigner dynamics, which is valuable for numerical simulations. Furthermore, the random walk model, including Lévy walks, underpins the superdiffusive behavior observed. Although we currently do not have a straightforward derivation from Eq. (13), we believe the random walk picture is essential for its physical meaning and numerical utility. We will clarify this rationale in the revised manuscript. 5) The reviewer is correct that there is no apparent restriction on the locality of the quasi-particles; they can be as far apart as necessary, provided they are not global operators, which would complicate coarse-graining. We will add further comments to discuss the implications of the structure in Fourier space and the locality of the dephasing jump operators.

---

## Round 1 · Referee Report · Anonymous (Referee 2) · 2024-9-14

Report

Dear Editor,

The paper by Yu-Peng Wang and collaborators discusses the transport properties of a dissipative chain of fermions in one and higher dimensions.
The paper is original, and while the techniques used are not innovative, the results are robust and quite surprising.

The explanation seems to be the choice of the Lindblad operators, with their modulated structure. Somehow, this seems similar to the study [https://journals.aps.org/prb/abstract/10.1103/PhysRevB.102.100301], which could be useful to cite.
Regarding anomalous transport with disorder, some results were presented in [https://journals.aps.org/prb/abstract/10.1103/PhysRevB.103.184202,https://scipost.org/SciPostPhys.12.6.189] which could be included among the outlook works [60,61].

Apart from these minor comments, I suggest the paper for publication in Scipost Physics.

Recommendation

Publish (surpasses expectations and criteria for this Journal; among top 10%)

  • validity: high
  • significance: high
  • originality: good
  • clarity: top
  • formatting: excellent
  • grammar: excellent

Author:  Jie Ren  on 2024-10-07  [id 4844]

(in reply to Report 4 on 2024-09-14)

We thank the reviewer for careful reading and for the recomendation.

We appreciate the reviewer for bringing up the relevant references. We have now included these citations in the revised version of the paper.

---

## Round 2 · Referee Report · Anonymous (Referee 5) · 2024-10-28

Report

The authors have addressed changes requested, so I am fine with publication.

Recommendation

Publish (meets expectations and criteria for this Journal)

---

## Round 2 · Author Response

Dear Editor,

We are pleased to resubmit our revised manuscript. We have carefully addressed the reviewers' comments and suggestions, which we believe have improved the clarity and robustness of the manuscript. We appreciate the reviewers’ valuable feedback and the opportunity to improve our work, and we hope that the revised version meets the expectations for publication.

Thank you for your consideration.

Sincerely,
Jie Ren
On behalf of all co-authors

---

## Round 2 · List of Changes

1. We have now included citations mentioned by reviewers.
2. We change the claim (line 93-94) that "the scaling function does not conform to the KPZ universality class due to a ballistic wavefront." to "the scaling function does not conform to the KPZ universality class.".
3. We replace the word "hydrodynamics" with "Wigner function dynamics" throughout the revised manuscript to avoid any confusion.
4. We have added comments in the main text (now in line 117) to clarify the point as reviewer suggested.

---

## Editorial Decision

published